# Responsive Supramolecular Polymers for Diagnosis and Treatment

**DOI:** 10.3390/ijms25074077

**Published:** 2024-04-06

**Authors:** Mónica Martínez-Orts, Silvia Pujals

**Affiliations:** Department of Biological Chemistry, Institute for Advanced Chemistry of Catalonia (IQAC-CSIC), 08034 Barcelona, Spain; monica.martinez@iqac.csic.es

**Keywords:** stimuli responsiveness, supramolecular polymers, non-covalent interactions, nanomedicine, stimuli-triggered delivery

## Abstract

Stimuli-responsive supramolecular polymers are ordered nanosized materials that are held together by non-covalent interactions (hydrogen-bonding, metal-ligand coordination, π-stacking and, host–guest interactions) and can reversibly undergo self-assembly. Their non-covalent nature endows supramolecular polymers with the ability to respond to external stimuli (temperature, light, ultrasound, electric/magnetic field) or environmental changes (temperature, pH, redox potential, enzyme activity), making them attractive candidates for a variety of biomedical applications. To date, supramolecular research has largely evolved in the development of smart water-soluble self-assemblies with the aim of mimicking the biological function of natural supramolecular systems. Indeed, there is a wide variety of synthetic biomaterials formulated with responsiveness to control and trigger, or not to trigger, aqueous self-assembly. The design of responsive supramolecular polymers ranges from the use of hydrophobic cores (i.e., benzene-1,3,5-tricarboxamide) to the introduction of macrocyclic hosts (i.e., cyclodextrins). In this review, we summarize the most relevant advances achieved in the design of stimuli-responsive supramolecular systems used to control transport and release of both diagnosis agents and therapeutic drugs in order to prevent, diagnose, and treat human diseases.

## 1. Introduction

Supramolecular chemistry, also referred to as “chemistry beyond the molecule” [1,2], studies the function and structure of supramolecular entities, i.e., supermolecules arising from the intermolecular binding between substrate species and macromolecular receptors [3]. In contrast to traditional chemistry, supramolecular science focuses on the non-covalent and reversible interactions between molecules. Since Lehn, Pedersen, and Cram were awarded the Nobel Prize in Chemistry (1987) “for their development and use of molecules with structure-specific interactions of high selectivity”, supramolecular research has steadily grown [4]. The combination of supramolecular chemistry and polymer science has given rise to a promising class of nanomaterials named supramolecular polymers. Unlike classical (covalent) polymers, supramolecular polymers are ordered self-assembled nanostructures that are built by non-covalent bridging of monomeric units [5].

The diverse applications of supramolecular polymers ranges from electronics to medicine [6]. The first use of supramolecular polymers in the marketplace was probably their application as coatings for heat-sensitive substrates. Due to their relatively low melting points, supramolecular polymers resist the high temperatures required in lithographic printing processes [7,8]. The differences in phase transition also allow the application of supramolecular polymers in ink-jet printing [9,10]. Moreover, the dynamic and responsive nature of supramolecular polymers endows them with excellent properties, such as self-healing and stimuli-responsiveness, for their application as reversible adhesive materials [11,12]. Besides their use in fabrics, supramolecular polymers are also attractive for cosmetics [13,14] and personal-care materials [15]. Supramolecular polymers also display electronic functionalities due to the planar orientation of the π-conjugated building blocks that stabilizes the 1D crystal [16,17]. The self-assembly of π-π stacked polymers yields well-defined nanostructures with efficient electrical properties such as good stability and high field-effect mobility [18].

The vast majority of biological recognition events take place in water. Thus, one of the major challenges in chemistry is to develop synthetic nature-based biomolecules that assemble in aqueous environments [19]. The huge family of supramolecular materials lies in the diversity of building blocks that assemble to form stable aggregates such as nanofibers, micelles, vesicles, and other nanostructures that are held together by weak interactions. The water-soluble and the non-covalent nature of supramolecular polymers, and thus their reversible self-assembly equilibrium, makes them attractive candidates for the design of biomaterials with medical applications. The use of supramolecular polymers for intracellular protein delivery, bone regeneration scaffolds, and drug delivery, among other biomedical therapies, lies on their versatility and the possibility of modulating their physical and mechanical properties [20]. Strikingly, the application of supramolecular amphiphilic scaffolds in regenerative medicine reveals the promising biomedical applications of these well-ordered systems [21,22,23,24,25,26].

The development of new strategies for the design, synthesis, and characterization of supramolecular polymers enables their application as potential nanocarriers for the delivery of drugs and diagnostic imaging contrast agents [20,27]. Nanomedicine is an emerging research area that applies nanotechnology to medical science and focuses on exploring key strategies for the diagnosis, treatment, and prevention of diseases at the molecular level [28,29,30]. During the last decades, much of the biomedical sector has worked on the development of new nanomaterials and their application to drug delivery, diagnosis, and therapeutics [31]. The clinical applications of nanomedicine includes the use of biocompatible nanomaterials in anti-inflammatory therapies for the treatment of cardiovascular disorders [32,33], in antiviral and antimicrobial drug delivery to fight infections [34,35], and in the development of nanoformulations effective against neurodegenerative disorders [36,37,38,39], among others.

Further advances in medicine and nanotechnology lead to the development of challenging approaches that aim to overcome the limits of traditional cancer therapeutics [40,41,42,43,44]. Nanomedicine-based techniques are used not only in clinical diagnosis for tumor detection, but also in the formulation and release of drugs with anticancer activity. In contrast to traditional chemotherapy strategies, the use of nanomaterials as drug delivery carriers provides numerous advantages in medicine: enhanced selectivity, stability and water solubility, extended half-life times, reduced side effects, and improved antitumor activity [45]. Polymers and nanoparticles play an increasingly important role in controlled release of drugs [46,47,48,49].

In this review, we will describe the characteristics of supramolecular polymers that make them attractive for biomedical use (Figure 1). We will discuss the most relevant discoveries in the design of outstanding responsive self-assemblies and their promising applications for the diagnosis and treatment of serious human diseases, especially cancer. Furthermore, we will emphasize the future goals in the field of supramolecular chemistry in relation to the design and synthesis of novel responsive nanodevices.

## 2. Supramolecular Self-Assembly: From Single to Complex Well-Ordered Structures

Molecular self-assembly is ubiquitous in nature and it covers a wide range of biological systems [50]. At the nanometer scale, there are various supramolecular systems with key biological functions (including amyloid fibrils, actin filaments, and microtubules) that are found in almost all living beings [51]. Supramolecular polymers are linear one-dimensional (1D) fiber-like assemblies that result from the spontaneous aggregation of monomeric units via non-covalent interactions [52]. Further reorganization of supramolecular polymers leads to high ordered structures such as supramolecular hydrogels, which are 3D scaffolds containing a network of nanofibers [53] (Figure 2). At this point, it is worth mentioning that 3D hydrogels are attracting special attention for their versatile biomedical applications related to their attractive properties such as their biocompatibility and controlled drug release [54,55].The engineering of supramolecular self-assemblies in aqueous media is guaranteed by the dual nature of amphiphiles [56,57]. Commonly, water-soluble supramolecular assemblies derive from amphiphilic monomers that are composed by covalently bonded hydrophobic and hydrophilic structural units [57]. The amphiphilic character of supramolecular building-blocks also mediates in the molecular self-assembly of one to three dimensional supramolecular polymers.

The weak interactions that hold natural self-assemblies together (i.e., the base pairing of DNA double-helix or protein folding [58,59,60,61,62]) serve as a model for the design of synthetic self-assembled systems. As mentioned above, the reversible nature of supramolecular assemblies is based on the non-covalent association between the different molecular units. The various types of reversible interactions found in supramolecular assemblies include metal-ligand coordination, hydrogen bonding, π-π stacking, and host–guest interactions (Figure 2) [63,64]. The reversible association of natural systems, such as actin filaments or DNA, has inspired synthetic chemists to design synthetic self-assemblies [1,65].

**Figure 2 ijms-25-04077-f002:**
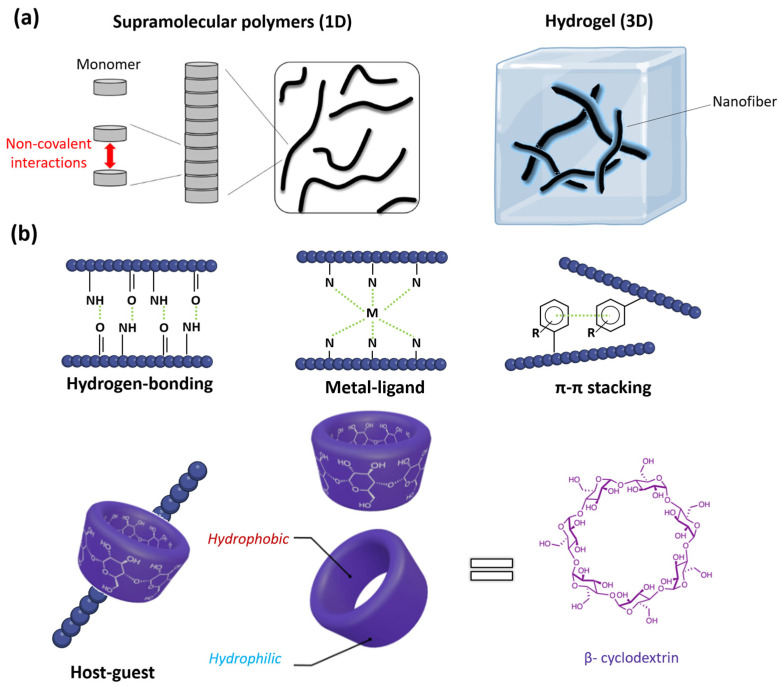
(**a**) Supramolecular polymers are fiber-like assemblies (left, figure adapted with permission from reference [52] under the terms of the Creative Commons CC BY license, accessed on 20 March 2024) that can further aggregate into high-ordered systems like supramolecular hydrogels (right). (**b**) The variety of supramolecular polymers arise from the diverse non-covalent interactions that facilitate the self-assembly process: hydrogen-bonding, metal-ligand coordination, π-π stacking, and host–guest interactions, an example of the self-assembly mediated with β-cyclodextrin is given (structure and the tridimensional view of the complexing host, adapted with permission from reference [66] under the terms of the Creative Commons CC BY license, accessed on 15 March 2024).

Inspired by the assembly of nucleobases, hydrogen bonds (HBs) are considered an attractive type of interaction that holds supramolecular polymers together to form well-defined aggregates [67]. For the design of supramolecular self-assemblies, it is necessary to pay carefully attention to the number and strength of HBs, pre-organization (to ensure the stability of hydrogen-bonded arrays), secondary electrostatic effect, and tautomerization. The high directionality and the diversity of HBs allow to formulate a plethora of hydrogen-bonded supramolecular polymers [63,67,68,69].

Moreover, metal–ligand coordination offers fascinating properties, such as tunable binding strength and excellent redox and photophysical behavior, for the design of attractive supramolecular materials. The potential application of metal–ligand complexes lies in their ability to adapt to the application of external stimuli [70]. In supramolecular chemistry, metal coordination has been extensively exploited for the design of different networks, including metallo-supramolecular polymers [71,72], to obtain functional nanostructured materials.

In chemistry, a common strategy in the design of water-soluble supramolecular polymers involves the use of a hydrophobic core (Figure 3) followed by the introduction of hydrophobic amino acid residues. The use of aromatics as central nuclei allows the versatile introduction of different functional groups, as well as the stability enhancement of supramolecular self-assemblies via π-stacking interactions [73]. 

In this context, the design of supramolecular polymers based on benzene-1,3,5-tricarboxamide (BTA) has been widely extended [74,75,76]. The structural features of BTA allow the molecular design of C3-symmetrical discotic (circular) amphiphiles that experiment spontaneous self-assembly in aqueous environments [77]. In the synthetic pathway, terminal, covalently attached hydrophilic side chains enhance water-solubility of BTA-based supramolecular helical fibers and reinforce their stability through HBs and π-π interactions [78,79] (Figure 3). During the last few years, triarylamines (TAA) have also been studied as attractive aromatic cores that can undergo supramolecular polymerization to provide versatile platforms [80]. In 2021, Picini et al. [81] synthesized a TTA-based macrocycle that was able to self-assemble into an electroactive 3D nanotube. Moreover, Oyspenko et al. published a “hot paper” in 2019 for their advances in the thermal control over the self-assembly of TTA supramolecular helical organogels that further stack into larger fibers [82]. More recently, Parida et al. [83] exploited the stabilizing interactions occurring in the supramolecular assemblies of a family of perylene-based polymers. In this context, perylene-bisimides (PBIs) had been previously reported to exhibit potent biological applications since they form stable self-assemblies in aqueous media [84].

The exploration of different aromatic nuclei in the construction of bio-attractive supramolecular polymers offers exciting avenues of research. For instance, porphyrin-based derivatives, owing attractive photophysical properties, are employed as highly fluorescent photosensitizers in photodynamic therapy. Upon light irradiation these molecules generate cytotoxic oxygen species, such as singlet oxygen (^1^O_2_), that can destroy tumor cells [85]. Liu et al. [86] reported a porphyrin-based supramolecular system with efficient use as a photosensitizer in the photodynamic therapy of bacterial invasion. In terms of responsiveness, the introduction of photosensitive azobenzene groups in the structure of porphyrin-based nanomaterials allowed Lu and co-workers [87] to obtain a multivalent amphiphile that exhibit dynamic and switchable self-assembly and optical properties in response to UV-Visible light irradiation. 

Among the various weak interactions that hold supramolecular polymers together, including triple hydrogen bonding of BTA-based assemblies, host-guest binding has attracted recent interest [88]. The design of host-guest supramolecular polymers includes a macrocyclic host that directly interacts with a specific guest moiety. Until the 1990s, guests involved in molecular recognition events have been limited to small molecules and simple ions. On the other hand, the use of cyclodextrins (CDs) (Figure 2) have gained attention in supramolecular chemistry during last few decades due to their attractive biomedical applications [89]. CDs are cyclic oligosaccharides composed of a hydrophilic outer layer, which enables water solubility, and an inner apolar cavity, which provides the appropriate matrix to properly encapsulate and carry hydrophobic drugs [90].

The combination of various non-covalent interactions constitute most orthogonal self-assembly strategies to obtain complexes with high specificity and selectivity [91]. Although the functionalization of hosts still remains a challenge, there are diverse macrocycles, besides cyclodextrins, with excellent properties. In this sense, pillar[n]arenes constitute a family of highlighting macrocycles that offer several advantages over other hosts [92]. These materials not only exhibit host-selective binding due to the symmetry and rigidity of their structures, but can also undergo easy functionalization on one or two positions or on all benzene rings, which confers them multi-responsiveness to different environmental stimuli. In addition, pillar[n]arenes are readily soluble in organic solvents. For all these reasons, pillar[n]arene-based supramolecular polymers have potential clinical applications, and they display an important role in cancer therapy [93,94,95,96].

## 3. Responsivity in Supramolecular Polymers

Due to their non-covalent nature, one of the most interesting properties of supramolecular polymers is their ability to assemble and disassemble in response to stimuli. This switchable behavior resembles the dynamic nature of the microtubule cytoskeleton, which can adapt to changes that may occur in the cell [97]. Supramolecular polymers that undergo changes in response to stimuli are unique materials that combine the dynamic properties of polymers, including long chains and directionality, with the ability to reversibly adjust their conformations as they are assembled through weak interactions [63]. 

To date, one of the greatest challenges in nanomedicine lies in the development of responsive supramolecular assemblies for controlled delivery. Numerous reviews [98,99,100,101,102] have reported the importance of using smart nanocarriers with stimuli-responsiveness for achieving selective release of therapeutic drugs or contrast agents. Controlled drug delivery enhances drug efficacy and avoids adverse (cytotoxic) effects in normal cells, since the loaded drugs are not released until they reach the injured site.

The various stimuli that can trigger reversible supramolecular self-assembly are classified as internal and external [103,104]. Some pathologies may alter the surrounding environment of injured tissues and induce dramatic changes in the physiological conditions of pH, redox, or enzyme expression (internal stimuli). On the other hand, light, ultrasound, and electric/magnetic field are classified as external stimuli. Furthermore, temperature is a versatile stimulus that can be applied externally (e.g., thermal annealing [105]) or act internally, as thermal variations occur in inflamed and tumor sites [106].

### 3.1. Temperature-Responsiveness

One of the most accessible properties of polymers is their responsiveness to temperature changes [103]. The ability to respond to thermal stimuli causes conformational changes in the polymer chains [107]. In water, temperature-responsive polymers undergo a thermally induced transition from the entirely dissolved coil to the globular aggregated form (Figure 4). The morphology of the assembled nanostructures depends significantly on the degree of hydrophilicity of a given block, as it can be altered with temperature [108]. The introduction of multiple H-bonding in the polymer structure is an effective strategy to tune the thermal-response and the morphology of amphiphilic copolymers [109].

### 3.2. Responsiveness to Internal Stimuli

#### 3.2.1. pH-Responsiveness

Supramolecular self-assembly can be regulated with protonation and deprotonation of pH-sensitive building blocks (i.e., histidine and lysine) [110,111,112]. This pH dependence is related to the electrostatic repulsion between two or more positively charged amino acid residues that are close together in space. Thus, the self-assembly of responsive supramolecular polymers can be controlled by modulating the pH of the surrounding medium [113,114].

#### 3.2.2. Redox-Responsiveness

There are other strategies in the design of stimuli-responsive assemblies, such as that related to redox responsiveness. Glutathione (ι-γ-glutamyl-ι-cysteinylglycine, GSH) is one of the most relevant agents of the cellular antioxidant defense [115]. The redox equilibrium between the thiol group (-SH) of GSH-cystein that is oxidized to form gluthathione disulfide (GSSG) governs the antioxidant function of glutathione. The introduction of disulfide bonds in the design of amphiphilic copolymers provides the redox-responsiveness. 

#### 3.2.3. Enzyme-Responsiveness

Among the stimuli-response related to temperature, pH, and redox balance, the responsiveness to enzymatic activity has also relevant interest. Specially in cancer, enzymes have attracted attention as vital markers in the therapeutic research [104]. Due to the specificity and selectivity of enzymes for unique substrates, enzyme-responsive assemblies can be design with the aim to detect phatological abnormalities. Specially, the fabrication of systems with responsiveness to proteases has special interest, since these enzymes are commonly overexpressed in cancer and other inflammation-related disorders [116,117,118]. This is the case of matrix metalloproteinases (MMPs) [119,120] and Cathepsin B. For instance, GPLGIAGQ (Gly-Pro-Leu-Gly-Ile-Ala-Gly-Gln) is an octapeptide that exhibits a high response to MMP2 and serves as inspiration to develop lipid-polymers sensitive to MMP2 activity [121]. On the other hand, the tetrapeptide sequence Gly-Phe-Leu-Gly (GFLG) is one of the most common motifs used in the design of polymeric systems with enzyme-responsiveness to Cathepsin B [122,123,124]. 

### 3.3. Responsiveness to External Stimuli

#### 3.3.1. Light-Responsiveness

The introduction of photo-responsive moieties in the synthetic pathway is a common approach used in the design of supramolecular amphiphiles sensitive to light [125,126,127,128]. Although there is a wide variety of photo-responsive molecules including stilbenes [129], diarylethenes [130] and spiropyrans [131], the use of azobenzenes has been widely emphasized during the last decade [79,132,133,134,135,136,137,138,139,140,141]. The N=N double bond of azo-compounds undergoes reversible photoisomerization from the stable trans isomer (E) to cis form (Z) under UV-light irradiation (Figure 5). The equilibrium returns to the initial configuration after irradiation with blue light or by thermal equilibration [142].

#### 3.3.2. Ultrasound-Responsiveness 

Besides its major limitation is their low drug-loaded content [143], ultrasound-responsive nanocarriers are highly investigated for its application in diagnostic imaging [144,145] and cancer treatment [146,147] due to its capability to enhance drugs permeability into cells [148,149], and thus increased delivery efficiency, in a minimally invasive manner [150]. Thermal and mechanical biological effects induced with ultrasound lead to biomedical applications related to the release of the loaded drug from the responsive nanocarrier [151]. Mechanical effects are directly related to cavitation, that is, ultrasound-induced oscillation and collapse of the cavities or bubbles arising from ultrasound sonoporation. On the other hand, thermal effects are due to the absorption of ultrasound energy by biological tissues. The temperature of the self-assembly polymerization and the solvents are two key factors that regulate the thermodynamic equilibrium (between metastable or stable states) that is directly related to the ultrasound responsiveness [152]. 

#### 3.3.3. Electrical-Responsiveness

Electro-responsive systems made of conducting polymers are particularly interesting. In contrast to light, ultrasound, or magnetic signals, electric stimuli are easy to generate and control [153]. A common strategy in the design of electro-responsive delivery systems is based on the introduction of intrinsically molecules able to undergo a redox reaction in the presence of an external electric field. In this sense, conducting polymers, such as poly(pyrrole) [154,155], can be used to complex negatively-charged drugs that are further released by the reduction of the conducting polymer [156]. In addition, the introduction of conducting polymers in the design of electro-responsive films also allows us to incorporate charged drugs during the formation of such films that can be used as coatings for neural applications [157]. The application of electric voltages enhances control drug delivery from electrical-responsive systems. 

#### 3.3.4. Magnetic-Responsiveness

To date, the introduction of magnetic targeting in the polymer science has considerably evolved. Harries et al. demonstrated that the use of hydrophilic block copolymers serves as a coat for stabilizing magnetic nanoparticles and enables their dispersion in biological fluids [158]. Despite the outstanding bioapplications of magnetic-responsive nanoparticles, the development of nanomagnetic organometalic copolymers have gained attention during the last decade [159]. In the design of magnetic polymers, it highlights the formulation of organic π-conjugated polymer magnets that exhibit interesting magnetic properties at low temperatures (below 10 K).

In addition, diverse methodologies have been developed to properly incorporate inorganic elements into self-assembled block copolymers [160,161,162]. For instance, cobalt-containing polymeric thin films, resulting from solvent annealing-induced self-assembly, display interesting ferromagnetic properties [163]. In fact, the incorporation of cobalt into self-assembled polymeric nanostructures facilitates magnetic targeting [164] and renders them potential candidates for magnetic imaging theranostic [165]. 

## 4. Biomedical Applications of Responsive Supramolecular Polymers 

The marvelous ability of supramolecular polymers to assemble/disassemble in response to external stimuli makes them potential candidates for biomedical applications. Supramolecular chemistry has emerged as a novel technology that offers numerous advantages in the context of diagnosis and treatment of cancer [166]. The exploitation of new drug delivery strategies based on supramolecular chemistry is one of the most challenging goals in chemotherapy [45,167]. 

In addition, the reversible and highly directional non-covalent interactions of supramolecular polymers make them attractive candidates for their use in regenerative medicine to support, guide, and stimulate regeneration of dysfunctional or even lost tissues [27]. The design of synthetic polymers that mimic the dynamic behavior of natural fibers has recently attracted considerable interest [168,169]. The eukaryotic cytoskeleton is composed of key supramolecular biopolymers, including actin filaments and microtubules, that are important for cell structure and function [51]. In particular, cellular microtubules are dynamic tubulin-based assemblies involved in the regulation of cell differentiation processes [170,171]. In the early 21st century, pioneering applications of bioactive peptide-based supramolecular nanofibers in regenerative medicine involve their potential use for tissue engineering of nerves [172], growth of blood vessels [173], and neural cell differentiation [174].

The use of stimuli-responsive nanosystems offers numerous advantages, including the versatility of shapes and structures that can be exploited for the design of novel molecules capable of evading the defense barriers they face when administered into the human body. The exploration of such smart nanosystems includes diagnosis sensors able to detect physical variables when used as imaging probes. In therapy, their application ranges from nanodevices for transport and release of therapeutic agents to smart surfaces for cell growth or tissue engineering [175]. 

### 4.1. Stimuli-Responsiveness in the Biological Environment

Especially in the fields of treatment and diagnosis, the impact of responsive supramolecular nanocarriers designed for stimuli triggered delivery is growing.

#### 4.1.1. Response to Internal Stimuli

In recent years, the development of redox-responsive supramolecular polymers has gained special interest, possibly due to the fact that the redox equilibrium can be altered in the presence of severe pathologies [176]. In nature, cellular homeostasis is regulated by the action of key reactive species, such as Glutathione (GSH), that maintain the physiologic redox balance. GSH is a highly exploited tumor marker, as elevated concentrations have been reported in tumor tissues compared to healthy tissues [177,178]. In 2018, Yu et al. [179] were pioneers in the development of a theranostic supramolecular polymer using β-cyclodextrin (β-CD) as a host segment linked to an anticancer drug (camptothecin) with a disulfide bond. In this work, they reported that drug release could be triggered by the elevated concentration of GSH inside tumor cells (Figure 6). The redox-responsiveness of this cleavable linker allow not only to maintain the anticancer efficacy toward the tumor site, but also to reduce the side effects on healthy cells. 

Moreover, redox-responsive supramolecular polymers containing ferrocenyl moieties is gaining increasing interest for controlled release [180]. In 2016, Zuo et al. [181] formulated a dual-redox-sensitive β-CD-ferrocene supramolecular amphiphilic polymer that self-assembles into supramolecular micelles and vesicles in aqueous solution. This system is able to trigger anticancer drug delivery in response to the elevated concentration of reactive oxygen species and GSH found in tumor cells. In 2020, Yin-Ku et al. [182] demonstrated the controlled release of doxorubicin (an effective chemotherapeutic drug [183]) that was entrapped in β-CD-ferrocene supramolecular micelles. In this case, both UV irradiation and redox response to hydrogen peroxide induce the dissociation of the micellar self-assembly and inhibit the proliferation of HeLa cells. 

Other multiple anomalies occurring in tumor microenvironment serve as internal stimuli for controlled delivery. This is the case of *temperature* rise [184], that allows for the effective transport and triggered release of therapeutic agents through the use of thermal-responsive nanocarriers [185]. In this field, the work developed by Kataoka, K. and coworkers [186] proved the use of an amphiphilic poly(oxazoline) as a molecular thermal switch to promote gen transfection efficiency. In this work, the thermoswitchable poly(oxazoline) was introduced between the hydprohilic shell of polyplex micelles and a cationic polypeptide segment. The resultant hydrophobic palisade protects the pDNA from nucleases and polyions that promotes effective nucleic acid delivery.

Also in cancer, unlike the physiological *pH* conditions of normal tissues (pH 7.4), the extracellular microenvironment of tumor tissues shows a slightly acidic pH (~6.5) [187]. This fact allows us to design pH-responsive nanoassemblies that can be used as drug targets for cancer therapy [188]. Furthermore, recent works [189,190] have demonstrated the potential applications of pH responsiveness for effective intracellular delivery of messengerRNA (mRNA). Although in the field of supramolecular polymers, the delivery of mRNA remains a challenge, there are few works [191,192] dedicated to improve the formulation of stimuli-responsive assembled micelles (that are highly evaluated for their use in human clinical studies [193,194]) to improve their stability for efficient gene transfection. In addition, the activity of *enzymes* associated with cancer [195] or inflammation-associated disorders [196] can be altered in such human pathologies. Although the development of enzyme-responsive supramolecular polymers remains a challenge, the use of supramolecular drug delivery systems that respond to the altered activity of matrix metalloproteases (MMPs) would be a promising approach to treat inflammatory-related illnesses, including cancer or rheumatoid arthritis disease [197]. Besides MMPs, lysosomal cathepsin B has been an enzymatic target of great interest in therapeutic trials. In cancer, this intracellular protease exhibits high levels of expression, which often induce secretion of Cathepsin B from cancer cells at the invasive edges of tumors [198,199]. 

During the exhaustive search for recent research on the design of supramolecular polymers that only respond to temperature, pH, or enzymes, work related to biomedical applications is lacking. However, the design of supramolecular polymers that respond to multiple stimuli is gaining particular interest in the last few years. For this reason, recent discoveries of supramolecular polymers that respond to thermal, pH, and enzymatic stimuli are collected in the following Section dedicated specifically to multi-responsiveness.

#### 4.1.2. Response to External Stimuli

Besides the environmental changes occurring in the human body in the presence of diverse pathologies, the external application of energy sources is also outstanding. As mentioned above, during the last few years, the design of multi-responsive supramolecular polymers is gaining interest. The introduction of structural motifs that respond to external stimuli, such as light, ultrasound, and electric/magnetic field, is commonly combined with the use of other diverse stimuli-responsive blocks to construct multi-responsive systems. However, in this Section, we would like to emphasize some relevant aspects that must be evaluated when external stimuli are applied in the clinic.

*Light* has been widely used as external stimuli to control supramolecular assembly in aqueous environments [128,200,201,202,203,204]. Photosensitive supramolecular assemblies can be used as nanotargets for drug delivery via activation with UV-Visible (<700 nm) or near-infrared (700–1000 nm) light depending on the characteristics of the release location and mostly depending on the photoswitch [205]. The use of light as minimally invasive stimuli provides high spatiotemporal resolution [206]. The incorporation of photo-responsive scaffolds in the design of supramolecular polymers as delivery systems allow us to control the assembly/disassembly of these nanostructures upon light irradiation. UV (ultraviolet) and visible light depths are restricted to hundreds of microns, whereas tissue penetration of NIR (near infrared) is greatest, up to a few centimeters, since water in tissues absorbs IR light [207,208,209,210]. For these reasons, NIR light is most appropriated for clinical applications and, moreover, it is less harmful to cells than other light sources.

Due to the advantages of *ultrasound* (millimeter precision [211], minimally invasive [212], intrinsic tissue penetration [104]), it has a potential stimuli for both therapeutic and diagnostic tools. Ultrasound-mediated delivery can be controlled with cavitation or thermal changes [213]. In the first case, cell-membrane permeability increases via sonoporation process which consists of the alteration of microbubbles (that can either stably oscillate or grow and violently collapse [214]) by the application of ultrasound frequencies [215]. On the contrary, ultrasound oscillations can be transferred to thermal energy, causing cell hyperthermia, which enhances the permeability of the cell [216]. The frequency of ultrasound (between kHz and MHz) depends on both the tissue and the organism, as well as on whether it is used for therapeutic or diagnostic purposes [214,217]. In fact, among the diverse uses of ultrasound treatments, it highlights their application for in vivo imaging and physiotherapy [218,219].

*Magnetism* is considered one of the best non-invasive external stimuli due to the facile tissue penetration [220] and the real-time response [106]. The application of an external magnetic field is widely used for medical imaging (magnetic resonance imaging, MRI). In the design of magnetic polymers, and it highlights the formulation of organic π-conjugated polymer magnets that exhibit interesting magnetic properties at low temperatures (below 10 K). In addition, diverse methodologies have been developed to properly incorporate inorganic elements into self-assembled copolymers [160,161,162]. For instance, cobalt-containing polymeric thin films, resulting from solvent annealing-induced self-assembly, display interesting ferromagnetic properties [163]. In fact, the incorporation of cobalt into self-assembled polymeric nanostructures facilitates the magnetic targeting [164] and renders them potential candidates for magnetic imaging theranostic [165]. 

Although the clinical application of electric-responsive supramolecular polymers is not yet achieved [104], external *electric fields* could also be potential stimulus to trigger drug release. However, the application of external stimuli is limited, since it can cause tissue injury and death [221]. In addition, electrical penetration depth is limited by human body barriers, such as skin, that attenuate the external electric field [222,223]. For these reasons, the irradiation parameters (voltage intensity and irradiation time) must be carefully selected [224].

One of the most important goals in the use of sensitive supramolecular systems in vivo is the formulation of self-assembled systems that respond to non-invasive or minimally invasive external stimuli. In 1978, Yatvin et al. reported for the first time the therapeutic effectiveness of a drug delivery system (DDS) for selective release on mild local hyperthermia [225]. However, traditional DDSs may display challenges related to side cytotoxic side effects. In this sense, responsive supramolecular self-assemblies are considered potential nanocarrier systems for effective delivery of drugs to overcome conventional DDSs limitations.

### 4.2. Multi-Responsiveness in the Biological Environment

During the last decades, various studies have focused on the development of new strategies to introduce stimuli-responsive motifs in the design of polymeric amphiphiles. These systems exhibit key structural features suitable for self-assembly in aqueous media, just as aforementioned. However, the use of smart supramolecular systems that respond to a single stimulus may be insufficient to achieve the desired control of the polymerization-depolymerization process [226,227]. The advantage of supramolecular multi-stimuli responsive carriers lies in the selective disassembly, and thus the controlled drug release, enhanced by the ability to respond to two or more stimuli [228]. 

Multi-responsiveness of supramolecular polymers offers novel strategies in the design of smart systems for bioapplications. To improve their selectivity in complex cellular environments, multi-responsive materials could be assembled/disassembled only when two different stimuli are applied simultaneously [140]. 

Although the design of multi-responsive supramolecular polymers (MRSPs) remains a challenge, several studies have been published in recent years. In this Section, some relevant examples in this area are listed below, following a chronological order of publication.

In 2016, Chen et al. [229] designed two supramolecular polymers that self-assemble into nanofibers and are sensitive to changes in *ionic strength*, *pH*, and *temperature*. The design of these novel polymers includes benzo-21-crown-7 (B21C7) host units that reversibly complexate with secondary ammonium salt via host-guest interactions. The reversible association of these MRSPs (Figure 7) was triggered by stimuli-response to *potassium cation* (K^+^), *chloride anion* (Cl^−^), *pH* (triethylamine/trifluoroacetic acid), and *temperature* (293–323 K).

In the exploration of new methodologies for the design of responsive supramolecular polymers, the use of cyclodextrin hosts is also promising. The complexation between CDs and guest stimuli-responsive molecules is an outstanding approach in the covalent formulation of supramolecular complexes [230]. Several studies have pointed out that CD-based host-guest derivatives composed of sensitive guests, such as azobenzene-containing polymers, could aggregate into supramolecular assemblies that undergo structural variations in response to one, two, or more stimuli [231]. Notably, in 2017, Li and co-workers developed a triple-stimuli-responsive amphiphile that exhibits dynamic β-cyclodextrin/azobenzene complexation and was able to self-assemble/disassemble into spherical micelles in response to *UV light*, *temperature*, and *redox potential* (Figure 7) [232].

In 2018, Zhou et al. [233] formulated a polymeric structure from β-cyclodextrin-poly(N-isopropylacrylamide) (β-CD-PNIPAM) and benzimidazole-terminated poly(ε-caprolactone) that self-assemble into supramolecular micelles (Figure 8). This reversible host–guest interaction enables the use of these smart nanocarriers with efficient release of anticancer drugs, and indeed high anti-cancer activity, at acidic pH values at 37 °C (*temperature* and *pH* triggered assembly).

In our research group, Fuentes et al. reported, in 2020, the synthesis of a MRSP that displays supramolecular polymerization into aqueous nanofibers triggered by four different stimuli [140]. In this work, it is demonstrated that it is possible to modulate the self-assembly of a discotic amphiphile toward the responsiveness to *light*, *pH, ionic strength*, and *temperature* (Figure 9). The multi-responsiveness of this system allows us to modulate the self-assembly equilibrium of supramolecular nanofibers in water.

**Figure 7 ijms-25-04077-f007:**
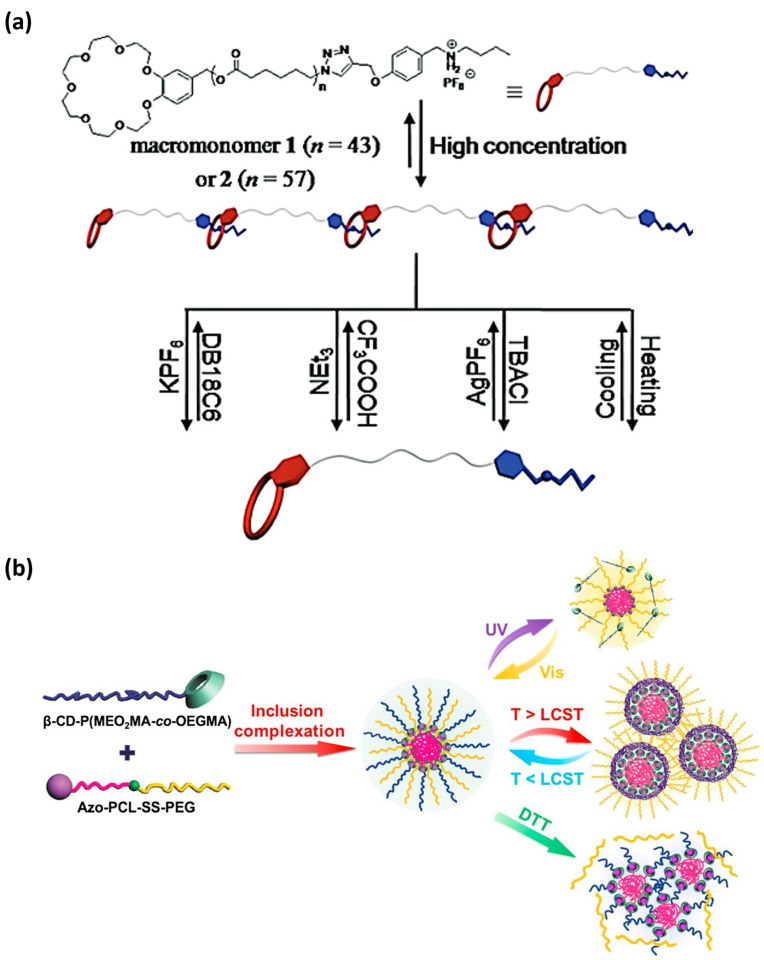
In this figure, two outstanding examples of three-responsive supramolecular polymers are highlighted. (**a**) Supramolecular self-assembly of heteroditopic macromonomers 1 and 2 (containing a shorter and a longer poly(ε-caprolactone) chain, respectively) is triggered by *ionic strength*, *pH*, and *temperature* variations (reprinted with permission from reference [229] under the terms of a License Agreement, order license ID 1454696-1, between the authors and Copyright Clearance Center (CCC), accessed on 27 February 2024) (**b**) The reversible supramolecular self-assembly of a multi-stimuli-sensitive supramolecular polymer (constructed by the host-guest complexation between β-cyclodextrin and azobencene) can be controlled by *light*, *temperature*, and *redox* changes (reprinted with permission from reference [232] under the terms and conditions provided by Elsevier and Copyright Clearance Center in agreement between the authors and Elsevier, license number 5737010923228, accessed on 27 February 2024).

**Figure 8 ijms-25-04077-f008:**
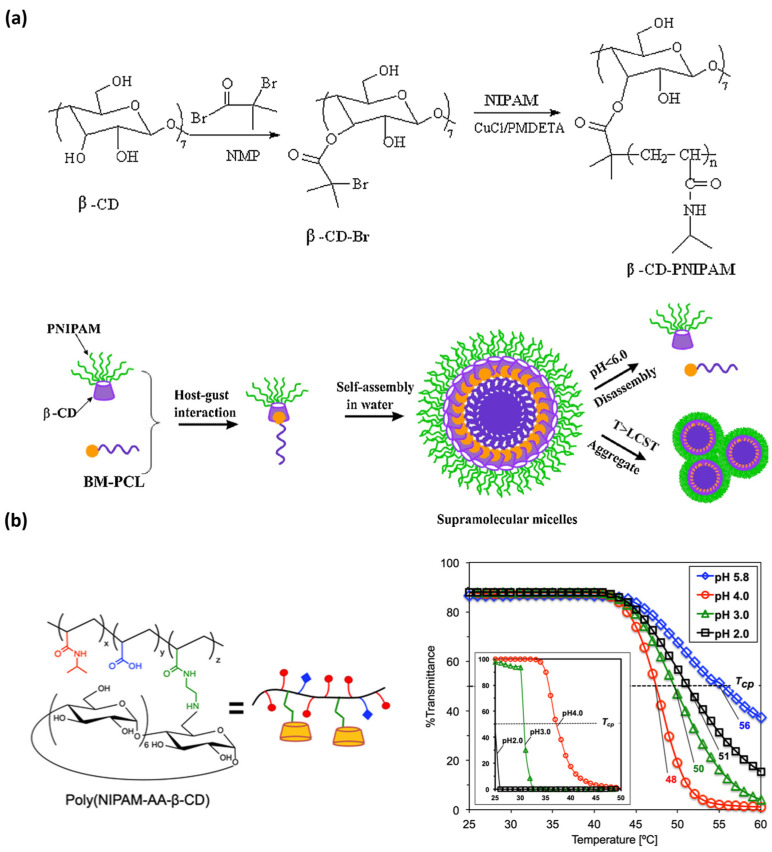
Two representative examples of recent supramolecular polymers sensitive to *temperature* and *pH* variations. (**a**) A pseudo-block copolymer (synthetized by coupling a thermos-responsive star polymer with a β-cyclodextrin core) self-assembles in water into dual responsive supramolecular micelles that respond to both *temperature* and *pH* (regarding the responsiveness of the host-guest complexation) variations (reprinted with permission from reference [233] under the terms and conditions provided by Elsevier and Copyright Clearance Center in agreement between the authors and Elsevier, license number 5736980493743, accessed on 27 February 2024). (**b**) Multi-stimuli responsive copolymer (left), poly(NIPAM-AA-β-CD), and its *thermo-* and *pH*-responsive behavior (right) (reprinted with permission from reference [234] under the terms of institutional subscription to American Chemical Society, accessed on 27 February 2024).

**Figure 9 ijms-25-04077-f009:**
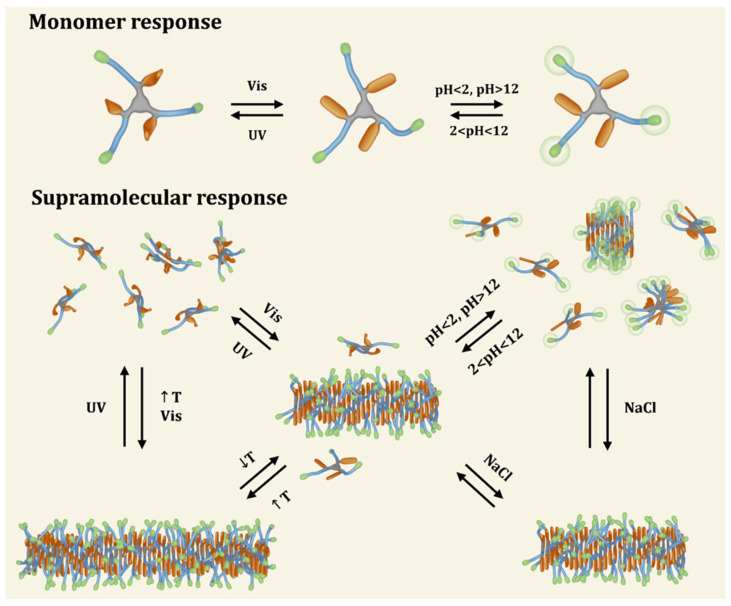
Schematic representation of a novel multi-responsive supramolecular polymer. Modular synthesis allowed to incorporate moieties that drive multiple equilibria due to their ability to respond to light (non-natural azobenzene amino acid, L-phenylalanine-4’-azobenzene, represented in orange), temperature (octa(ethylene glycol) amino acid, highlighted in blue) and pH/ionic strength (C-terminal lysine, colored in green) stimuli (reprinted with permission from reference [140] under the terms of the Standard ACS AuthorChoice/Editors’ Choice usage agreement, accessed on 25 August 2023).

More recently, in 2021, Kawano et al. [234] designed a multiple-stimuli-responsive copolymer containing sensitive motifs with responsiveness to temperature and pH variations (Figure 8). The formulation of this MRSP was comprised of a poly(N-isopropylacrylamideco-acrylic) acid that was structurally modified with aminoethyl acrylamide-β-CD pendant groups as thermo- and pH-responsive motifs. Depending on the pH, the protonation of the charged surface (carboxylic acid, secondary amino group, and amide segment) of this MRSP promotes the formation of cationic or anionic species that regulates the intermolecular aggregation. In addition, they reported that the cloud point (Tcp) of poly (NIPAM-AA-β-CD) changes upon pH variation. They demonstrated that turbidity becomes higher by heating at increasing pH values (from 2.0 to 5.8). In this work, they also reach the molecular recognition of a lipophilic dye by the copolymer amphiphile. One of the most promising applications of poly (NIPAM-AA-β-CD) is, then, its use for lipophilic pharmaceutical drug encapsulation and further release by modulating pH.

The application of responsive supramolecular polymers for ultrasound imaging still remains a challenge due to the high intensity required for effective ultrasound responsiveness [213]. However, Wei et al. [235] have recently reported a strategy to design a dual-responsive polymersome with triggered self-assembly by response to ultrasound and pH. They combined poly(ethylene oxide) as a corona-forming block with poly(methoxyethyl methacrylate) as the ultrasound-responsive motif. This system provided promising results for doxorubicin drug release in both in vivo and in vitro experiments.

## 5. Future Perspectives

Polymers are ubiquitous in all living beings, and they play an essential role in cell growth and function. In nature, the weak but highly cooperative interactions that hold functional biopolymers together endow them with the unique property of undergoing reversible changes in response to external stimuli. In order to understand the external control mechanisms of biopolymers, many efforts have been focused on attempting to mimic this behavior in synthetic systems. At the end of the last century, the concept of ‘smart polymers’ emerged in attempts to understand the responsiveness of biopolymers by mimicking this behavior in synthetic systems. The potential applications of smart polymers in biotechnology and medicine are well known.

There is still an outstanding challenge in the *design* and *synthesis* of responsive supramolecular polymers: to precisely control the shape, size, and stability of the self-assembled aggregates from aqueous supramolecular polymerization events [51,236]. Fortunately, there are some examples in the literature that open new insights into the precise control of supramolecular polymers growth [237]. In addition, it is important to achieve precise control on the morphology of that responsive nanocarriers. In this sense, the ability to modulate the shape and size of supramolecular polymers would open perspectives in the design of specific systems for loading and transport specific drugs/contrast agents. On the other hand, it is also necessary to design responsive supramolecular polymers with good stability. When nanocarriers are administrated, they drive through bloodstream and resist adverse conditions. Therefore, new supramolecular polymers designed for therapeutic purposes must exhibit some degree of robustness in order to remain stable (against dilution effects or the action of blood proteins) until they reach the target [238]. Also, stimuli-responsiveness could be used to modulate the effect of protein corona formation, leveraging stimuli-responsive strategies to enhance stealth and therapeutic efficacy [239].

To date, supramolecular research has evolved significantly. The understanding of biological recognition processes is increasingly elucidated by the strikingly advances in engineering the functionality of smart natural systems. Several fields of study offer new avenues to address the design and synthesis of supramolecular polymers, with particular emphasis on the development of responsive self-assembled systems. The great biomedical impact of sensitive supramolecular polymers lies in their ability to respond to one, two, or more stimuli, which makes them suitable drug nanocarriers for the diagnosis, prevention, and treatment of serious human diseases. On the other hand, the promising use of responsive supramolecular assemblies in *diagnosis* has been also studied. The development of new nanodevices for diagnosis is essential to achieve early detection of functional abnormalities to prevent disease progression. For instance, in medical imaging, magnetic resonance imaging (MRI) offers attractive features, including the absence of radiation damage and the high contrast it provides between neurological, musculoskeletal, and other soft tissues [240]. In MRI, the use of paramagnetic transition metal ion chelates as contrast agents is highly exploited. Mainly Gd-DTPA (Gadolinium-diethylene-triaminepentaacetic acid)-based systems have provided information on intracranial lesions for a few decades now [241,242], with relevant use for in vivo detection of β-amyloid deposits in mouse models of Alzheimer’s Disease [243,244]. 

The possibility to control and tune the structure of supramolecular polymers makes them also attractive for their use as bioimaging probes. In the last decade, several studies revealed that self-assembled nanofibers functionalized with fluorescent groups will be potential agents for their use in bioimaging [245,246]. Although the application of supramolecular assemblies in diagnosis remains a challenge, some recent advances can provide insights [247]. Moreover, multi-stimuli responsive nanosystems are promising candidates in the development of new diagnostic tools. The possibility to use a supramolecular polymer responsive to both an external and an internal stimulus would be promising for acquiring two or more different imaging tests that require the patient to be anesthetized.

Besides therapy and diagnosis, advanced functionality through the integration of multiple functional components will endow nanomaterials with bioinspired capabilities, mimicking biological structures and functions, e.g., molecular recognition, self-healing.

## 6. Concluding Remarks

The weak association of natural assemblies has inspired the design of synthetic systems. The combination of supramolecular chemistry with polymer science has given rise to a novel class of nanoassemblies named supramolecular polymers. Unlike covalent systems, supramolecular polymers are well-ordered assemblies hold together by weak and non-covalent interactions. The main forces that drive supramolecular self-assembly are hydrogen-bonding, metal-ligand coordination, π-stacking, and host–guest interactions. In this review, we gave a brief overview of the various non-covalent interactions that hold supramolecular nanostructures together, as well as some outstanding strategies for the design of these systems. Specifically, we described a common strategy in the design of water-soluble supramolecular polymers which consists of the use of a hydrophobic core (such as benzene-1,3,5-tricarboxamine, perylene-bismide, triarylamine or porphyrin). Furthermore, we highlighted the most remarkable properties of supramolecular polymers that make them potential candidates for a wide variety of applications. These systems hold attractive features that make them potential candidates in biomedicine. The modular, tunable, and reversible self-assembly of supramolecular polymers confers them the attractive interest for achieving control in drug delivery. Especially, we focused this review on one of the most marvelous properties of supramolecular polymers: their ability to assemble and disassemble in response to stimuli. Firstly, we described the most relevant properties of the diverse internal (pH, redox potential, enzyme activity) and external (light, ultrasound, electric/magnetic field) stimuli. In the case of temperature, it is a versatile stimulus that can be applied as both internal or external stimulus. Later, we focused on the biomedical applications of responsive supramolecular polymers including drug delivery. In this regard, we have collected the most recent discoveries that represent major breakthroughs for supramolecular medicine. To date, the most relevant responsive supramolecular polymers have been designed with the ability to respond against *pH, temperature*, and *light.* Moreover, the most representative examples found in the literature gather supramolecular polymers based on host-guest complexation. Although the translation of responsive supramolecular polymers into the clinic still remains a challenge, in the literature we found outstanding works that will open new insights into this field. 

## Figures and Tables

**Figure 1 ijms-25-04077-f001:**
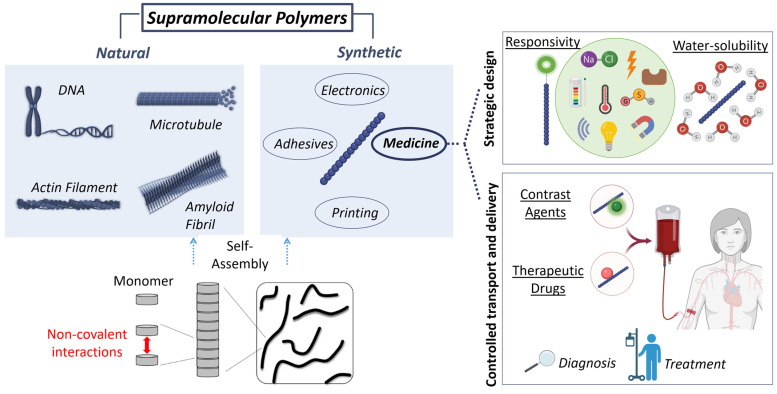
Supramolecular polymers are self-assembled systems held together by non-covalent interactions. DNA, microtubules, actin filaments, or amyloid fibrils are representative examples of natural supramolecular polymers that have inspired the design of synthetic assemblies. Among the diverse applications of supramolecular polymers, this review focuses on their biomedical use. For their application in medicine, the strategic design of supramolecular polymers includes two main concepts: responsivity, and water-solubility. Responsive supramolecular-polymers can be used to control transport and delivery of therapeutic drugs or contrast agents for their use in treatment or diagnosis, respectively.

**Figure 3 ijms-25-04077-f003:**
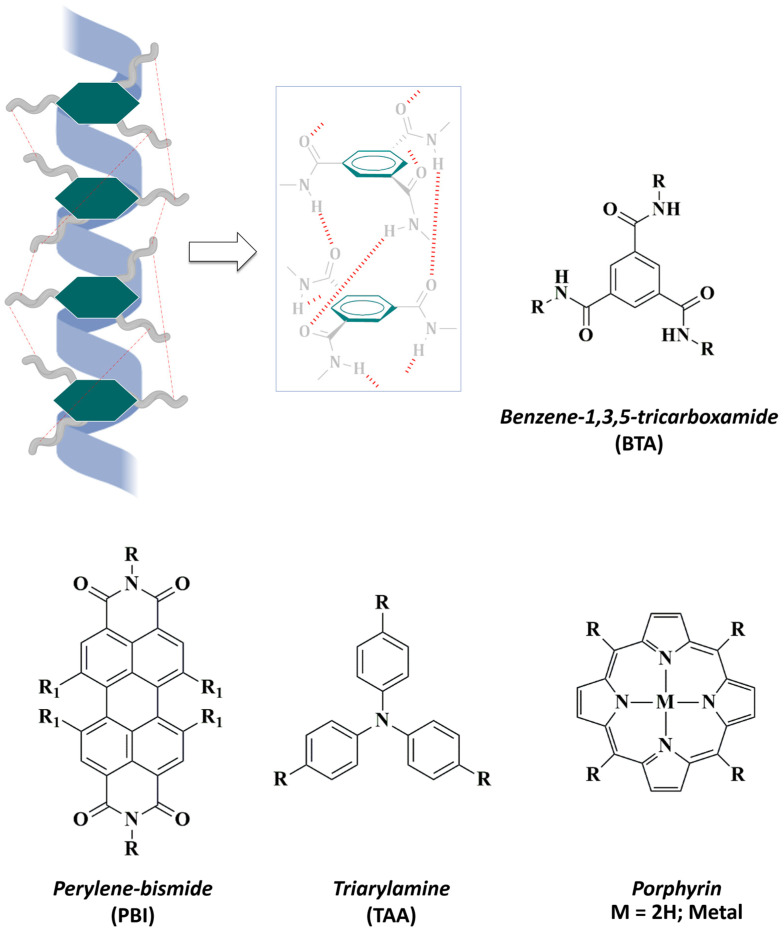
Hydrophobic aromatic cores commonly used in the design of water-soluble supramolecular polymers. (**Top**) Representative illustration of the columnar helical aggregates of SPs based on benzene-1,3,5-tricarboxamide (BTA): three-fold hydrogen bonding (red) stabilize the supramolecular assembling. (**Bottom**) Other well-known hydrophobic cores are: perylene-bismide (PBI), triarylamine (TAA), and porphyrins.

**Figure 4 ijms-25-04077-f004:**
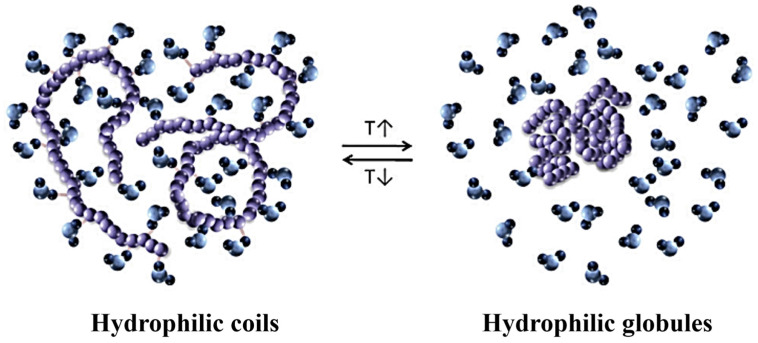
Thermally induced coil-to-globule transition, arising from the temperature dependence of the hydrophilic and hydrophobic interactions between the polymer (colored in purple) and water molecules (represented in blue shades) of the solvent (figure reprinted with permission from reference [107] under the terms of a License Agreement, order license ID 1454499-1, between the authors and Copyright Clearance Center (CCC), accessed on 26 February 2024).

**Figure 5 ijms-25-04077-f005:**
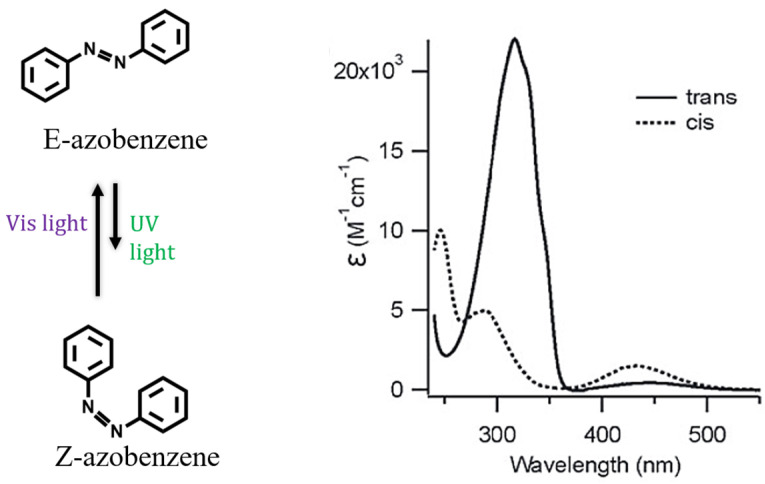
Azobenzene photoisomerization (**left**) and electronic absorption spectra of trans and cis isomers (**right**) that are, respectively, induced using irradiation with visible or UV light (reprinted with permission from reference [142] under the terms of a License Agreement, order license ID 1454683-1, between the authors and Copyright Clearance Center (CCC), accessed on 27 February 2024).

**Figure 6 ijms-25-04077-f006:**
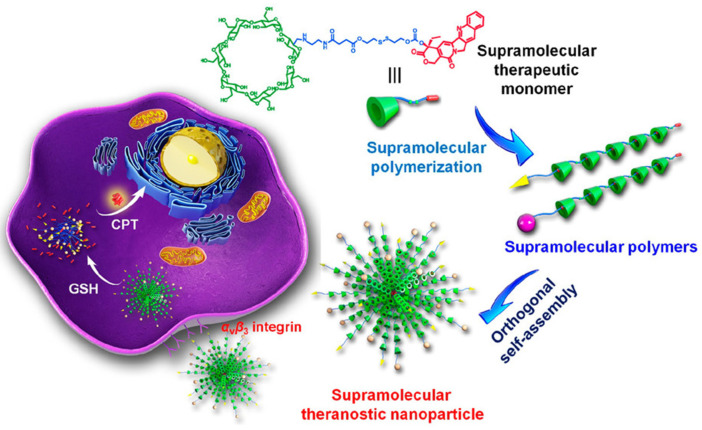
A novel supramolecular therapeutic polymer constructed from host–guest complexation between β-cyclodextrin (β-CD) and camptothecin (CPT) orthogonally self-assembles into supramolecular theranostic nanoparticles with anticancer functions. The glutathione (GSH) responsive capability of the nanoassemblies endow rapidly dissociation after cell internalization (reprinted with permission from reference [179] under the terms of institutional subscription to American Chemical Society, accessed on 27 Februrary 2024).

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
