# Peer review of "Responsive Supramolecular Polymers for Diagnosis and Treatment"

_ijms, 2024, doi:10.3390/ijms25074077_

Round 1

Reviewer 1 Report

Comments and Suggestions for Authors

1. At the current stage, this paper is a hodgepodge of fragments, to some extent.  Section 3 contains a huge amount of irrelevant information unrelated to the article's main topic. What's the meaning of introducing so general information on responsivity? Particularly, the examples in this section actually is not Supramolecular Polymer. This section should be significantly shortened. A simple table or figure is already enough.

2. I believe the authors want to focus on guest-host interaction. However, the paper still contains a huge amount of irrelevant information, such as Figure 10. Please check the manuscript and keep accuracy.

3. The authors should discuss in details that why the responsivity is needed (https://doi.org/10.1021/jacs.0c09029). 

Author Response

We want to thank the reviewer for the thorough evaluation of our manuscript. In particular, we want to thank his/her effort in widening our work, making the perspectives and interpretations of our review much more interesting. The modifications included in the original manuscript are described below.

  1. At the current stage, this paper is a hodgepodge of fragments, to some extent.  Section 3 contains a huge amount of irrelevant information unrelated to the article's main topic. What's the meaning of introducing so general information on responsivity? Particularly, the examples in this section actually is not Supramolecular Polymer. This section should be significantly shortened. A simple table or figure is already enough.

We thank the reviewer for his/her comment, we have tried to rewrite the paper in a more connected, story-like manner. Regarding section 3, we aim to describe in detail the different stimuli in this section, while explaining their application in the biological context in section 4. We agree with the reviewer that the section was too long, and we have summarized it as much as possible, removing the examples that were not on supramolecular polymers.

  1. I believe the authors want to focus on guest-host interaction. However, the paper still contains a huge amount of irrelevant information, such as Figure 10. Please check the manuscript and keep accuracy.

We agree with the reviewer and both Figure 10 and the related paragraph have been removed.

  1. The authors should discuss in details that why the responsivity is needed (https://doi.org/10.1021/jacs.0c09029). 

We thank the reviewer for raising this key point, thus we have added a paragraph (line 224) to detail the importance of responsivity, including the reference given.

Reviewer 2 Report

Comments and Suggestions for Authors

The authors have reported as follows:

Stimuli-responsive supramolecular polymers represent organized nanoscale materials held together by non-covalent interactions, capable of reversible self-assembly triggered by specific stimuli. Their non-covalent structure grants them the capacity to adapt to external cues or changes in the environment, rendering them appealing for diverse biomedical applications. Thus far, supramolecular research has predominantly focused on developing intelligent water-soluble self-assemblies to emulate the biological functions of natural supramolecular systems. Various synthetic biomaterials have been engineered with responsiveness to regulate and initiate, or suppress, aqueous self-assembly. A significant area of investigation in supramolecular chemistry involves crafting water-soluble supramolecular nanocarriers capable of disassembling in response to stimuli. In this overview, the authors consolidate the most notable advancements in designing stimuli-responsive supramolecular systems utilized for controlling the transport and release of both diagnostic agents and therapeutic drugs, aiming to prevent, diagnose, and treat severe human ailments.

However, the manuscript could benefit from additional discussion and data. Some of the points with the current manuscript are outlined below, and it is suggested that these be addressed in a major revision. Therefore, the reviewer recommends that the paper be reconsidered following these revisions.

Reviewer’s Feedback:

Abstract: The abstract should be revised to adopt a more scientific tone, prioritizing a comprehensive discussion of the substantial data presented in the manuscript. Emphasizing a succinct summary of key findings and their implications is crucial.

Figures: It is recommended to include an infographic as Figure 1 to encapsulate the entire content of the manuscript and visually represent the core concepts discussed. Figures 1-3 are deemed too small and should be amalgamated into a single figure. Additionally, multiple smaller figures throughout the manuscript should be merged.

Sections 3.3.2 and 3.3.3: These sections are considered insufficiently detailed and should be expanded with additional references from pertinent literature.

Introduction: The introduction warrants expansion to provide a comprehensive overview of the scientific issues addressed in this research. Incorporating references to recent scholarly works by esteemed authors such as Ramiro Manuel Velasco Delgadillo, Mohan G. Kalaskar, EV Barrera, N. Mamidi, and Javier Villela Castrejón is advisable to establish a robust foundation for discussing the various attributes and applications of supramolecular hydrogels and biomaterials. Particular emphasis should be placed on aspects such as biocompatibility, degradation kinetics, and controlled drug release, especially concerning their advantages in wound healing. Ensure appropriate citation of the recommended articles.

Clinical Applications/Trials: Introduce a dedicated table and section within the revised manuscript to present clinical applications and trials involving supramolecular materials/hydrogels. This will provide readers with valuable insights into the practical aspects of the current field.

Future Prospects: Although this section is well-supported with literature, it is suggested to minimize citations and provide original insights into the research field.

Conclusions: The conclusions section is deemed insufficiently detailed and should be expanded to provide a more comprehensive summary.

Please address these comments in your manuscript to enhance its scientific rigor and clarity.

Author Response

We want to thank the reviewer for the thorough evaluation of our manuscript. In particular, we want to thank his/her effort in widening our work, making the perspectives and interpretations of our review much more interesting.  The modifications included in the original manuscript are described below.

The authors have reported as follows:

Stimuli-responsive supramolecular polymers represent organized nanoscale materials held together by non-covalent interactions, capable of reversible self-assembly triggered by specific stimuli. Their non-covalent structure grants them the capacity to adapt to external cues or changes in the environment, rendering them appealing for diverse biomedical applications. Thus far, supramolecular research has predominantly focused on developing intelligent water-soluble self-assemblies to emulate the biological functions of natural supramolecular systems. Various synthetic biomaterials have been engineered with responsiveness to regulate and initiate, or suppress, aqueous self-assembly. A significant area of investigation in supramolecular chemistry involves crafting water-soluble supramolecular nanocarriers capable of disassembling in response to stimuli. In this overview, the authors consolidate the most notable advancements in designing stimuli-responsive supramolecular systems utilized for controlling the transport and release of both diagnostic agents and therapeutic drugs, aiming to prevent, diagnose, and treat severe human ailments.

However, the manuscript could benefit from additional discussion and data. Some of the points with the current manuscript are outlined below, and it is suggested that these be addressed in a major revision. Therefore, the reviewer recommends that the paper be reconsidered following these revisions.

Reviewer’s Feedback:

Abstract: The abstract should be revised to adopt a more scientific tone, prioritizing a comprehensive discussion of the substantial data presented in the manuscript. Emphasizing a succinct summary of key findings and their implications is crucial.

We have modified the abstract to provide a better summary of the review.

Figures: It is recommended to include an infographic as Figure 1 to encapsulate the entire content of the manuscript and visually represent the core concepts discussed. Figures 1-3 are deemed too small and should be amalgamated into a single figure. Additionally, multiple smaller figures throughout the manuscript should be merged.

We have prepared as suggested Figure 1, summarising all the content of the review, and it has been included in the manuscript. In addition, we have joined several figures into single ones. The new merged figures are Figures 2, 3, 7 and 8.

Sections 3.3.2 and 3.3.3: These sections are considered insufficiently detailed and should be expanded with additional references from pertinent literature.

We have expanded sections 3.3.2 and 3.3.3. to give more detail about ultrasound and electric-field stimuli.

Introduction: The introduction warrants expansion to provide a comprehensive overview of the scientific issues addressed in this research. Incorporating references to recent scholarly works by esteemed authors such as Ramiro Manuel Velasco Delgadillo, Mohan G. Kalaskar, EV Barrera, N. Mamidi, and Javier Villela Castrejón is advisable to establish a robust foundation for discussing the various attributes and applications of supramolecular hydrogels and biomaterials. Particular emphasis should be placed on aspects such as biocompatibility, degradation kinetics, and controlled drug release, especially concerning their advantages in wound healing. Ensure appropriate citation of the recommended articles.

We have introduced a reference from the recommended authors (Pharmaceuticals 2021, 14(4), 291; https://doi.org/10.3390/ph14040291; reference 54, line 119) and  added a sentence regarding the importance of hydrogels in biomedicine (from line 116). Research on hydrogels is of high interest, but it is out of the scope of the main topic of this review, supramolecular polymers. We have decided to add only this paragraph about this three-dimensional assemblies to keep the focus of the article on supramolecular polymers.

Clinical Applications/Trials: Introduce a dedicated table and section within the revised manuscript to present clinical applications and trials involving supramolecular materials/hydrogels. This will provide readers with valuable insights into the practical aspects of the current field.

Indeed, clinical applications should we added if supramolecular polymers had reached that stage. We thank the reviewer to suggest this modification, however, there is no clinical application for supramolecular polymers yet. As the reviewer comments, there are examples for supramolecular materials and hydrogels, but those are not the main focus of our review.

Future Prospects: Although this section is well-supported with literature, it is suggested to minimize citations and provide original insights into the research field.

In this section, several citations have been removed following the recommendation of the reviewer. In addition, some ideas have been included to provide original insights.

Conclusions: The conclusions section is deemed insufficiently detailed and should be expanded to provide a more comprehensive summary.

The section dedicated to conclusions has been considerably modified and expanded.

Please address these comments in your manuscript to enhance its scientific rigor and clarity.

Reviewer 3 Report

Comments and Suggestions for Authors

Mónica Martínez-Orts and Sílvia Pujals reviewed a topic entitled “Responsive Supramolecular Polymers for Diagnosis and Treatment. The review primarily focused on various engineering strategies and design criteria of bioresponsive supramolecular assemblies for the effective release and transport of diagnostic and therapeutic agents to diagnose, prevent, manage, and treat diseases. 

The manuscript can be accepted for possible publication in the International Journal of Molecular Sciences. However, we recommend that the authors carefully address the following major and minor comments and concerns raised by the reviewer to reach a broader audience in bioengineering, biology, and materials science before considering a possible publication.

1.     Footnotes are highly recommended below the Figures. A brief explanation of the figure is necessary for a quick understanding of the readers. For example, Figure 1 describes the formulation of polymeric micelle and polymersome. We recommend the authors describe how a block copolymer chooses to form a micelle or polymersome. A brief emphasis on the significant difference between a polymeric micelle and a polymersome is essential.

2.     The authors used an A-B-A type block copolymer. Show which is a hydrophilic unit and which is a hydrophobic unit. For example, the red-colored block is hydrophobic, and the blue is hydrophilic. In the polymeric micelle, where is the second blue-colored block hidden? Does the same A-B-A type block copolymer form a micelle and polymersome? If so, what factors determine the micelle formation and polymersome formation? A clear description is recommended. 

3.     The authors describe the mechanism of thermal switching of hydrophilic coils to hydrophobic globules in Figure 6. However, no relevant application-oriented examples were discussed. We encourage the team to discuss relevant application-oriented examples with the following information on nucleic acid delivery. For example, A thermoswitchable amphiphilic poly(2-n-propyl-2-oxazoline) (PnPrOx) was introduced between a hydrophilic poly(2-ethyl-2-oxazoline) block and a cationic poly(L-lysine) segment. The PnPrOx created a hydrophobic palisade due to the collapse of the PnPrOx segment above the lower critical solution temperature, resulting in a spatially aligned hydrophilic−hydrophobic double-protected polyion complex micelle loaded with plasmid DNA (pDNA). This hydrophobic palisade protected the pDNA from nuclease attacks and polyion exchange reactions, promoting effective gene transfection compared to micelle without hydrophobic palisade (Biomacromolecules. 2016;17(1):354-61).

4.     In the literature, we can find many reviews on stimuli-sensitive assemblies and their applications. How is this review significantly different from those existing reviews? An excellent review article is forward-looking and provides a framework for future direction for readers. 

5.     The authors discussed only the rationale for responsiveness to internal stimuli. However, examples were missing. We recommend the authors add essential applications for delivering therapeutic and diagnostic agents. For example, a thiol-responsive core crosslinking was introduced into the core compartment to prevent premature disassembly of nucleic acid therapeutics-loaded polyplex micelle, thereby promoting transfection efficiency. A pDNA-packaged crosslinked PEG-poly(L-lysine) (PEG-PLys) block copolymer-based polyplex micelle offered remarkable stability in the extracellular environment (10 µM glutathione), thereby enhancing cellular internalization at tumor site. After cytoplasmic entry, these crosslinks are cleaved in the reductive milieu in response to a highly reduced glutathione (10 mM) level, triggering the effective release of loaded pDNA (Biomaterials. 2014;35(20):5359-5368). Thiol core crosslinked micelles loading siRNA and mRNA effectively protected and delivered their nucleic acid cargoes into the cells (Macromol Rapid Commun 2022;43(12):e2100698J Drug Target 2019;27(5-6):670-680, and Macromol Rapid Commun 2016;37(11):924-33). 

6.     The authors discussed the rationale of pH-responsiveness. However, they failed to discuss the examples of pH-responsiveness in drug delivery. We recommend the authors incorporate the following discussion. MessengerRNA (mRNA) delivery has garnered spotlight attention as a versatile nucleic acid drug, not only for disease management and treatment but also for preventing infectious diseases (Proc Natl Acad Sci U S A2024;121(11):e2307800120). We recommend the authors introduce a few examples of stimuli-responsive polymers for effective intracellular mRNA delivery. For example, the delivery efficiency of mRNA delivery systems is greatly hampered by entrapment within endosomal/lysosomal compartments. To overcome this endosomal/lysosomal entrapment, an acidic pH-responsive negatively charged polymer (i.e., charge-conversion polymer) is coated onto a positively charged polyion complex. After reaching the endosomal/lysosomal compartment, followed by endocytosis, the coated charge-conversion polymer converts to a positively charged polymer from the original negative charge in the acidic pH milieu of endosomal/lysosomal compartments, facilitating smooth endosomal disruption. This design of charge-conversion polymer facilitated the efficient endosomal escape of the mRNA-packaged polyplex compared to a polyplex coated with a polymer lacking pH-responsiveness (Macromol Rapid Commun 2022;43(12):e2100754)

7.     Line 256: matrix metalloproteinases (MMPSs). The correct acronym and most widely used is MMPs, not MMPSs. 

8.     Lines 242 and 357: Glutathione (L–γ-glutamyl-L-cysteinylglycine, GSH). Use this amino acid description only at line 242. 

9.     Line 360: What is theranostic SP?

10.  Provide complete amino acids of codes used in GPLGIAGQ

11.  The manuscript mainly uses the previously published review articles as citations. We recommend the authors incorporate the original discussion and original article citations. 

Author Response

We want to thank the reviewer for the thorough evaluation of our manuscript. In particular, we want to thank his/her effort in widening our work, making the perspectives and interpretations of our review much more interesting. The modifications included in the original manuscript are described below.

Mónica Martínez-Orts and Sílvia Pujals reviewed a topic entitled “Responsive Supramolecular Polymers for Diagnosis and Treatment. The review primarily focused on various engineering strategies and design criteria of bioresponsive supramolecular assemblies for the effective release and transport of diagnostic and therapeutic agents to diagnose, prevent, manage, and treat diseases. 

The manuscript can be accepted for possible publication in the International Journal of Molecular Sciences. However, we recommend that the authors carefully address the following major and minor comments and concerns raised by the reviewer to reach a broader audience in bioengineering, biology, and materials science before considering a possible publication.

  1. Footnotes are highly recommended below the Figures. A brief explanation of the figure is necessary for a quick understanding of the readers. For example, Figure 1 describes the formulation of polymeric micelle and polymersome. We recommend the authors describe how a block copolymer chooses to form a micelle or polymersome. A brief emphasis on the significant difference between a polymeric micelle and a polymersome is essential.

We thank the reviewer for this suggestion and figure captions have been expanded in all cases.

  1. The authors used an A-B-A type block copolymer. Show which is a hydrophilic unit and which is a hydrophobic unit. For example, the red-colored block is hydrophobic, and the blue is hydrophilic. In the polymeric micelle, where is the second blue-colored block hidden? Does the same A-B-A type block copolymer form a micelle and polymersome? If so, what factors determine the micelle formation and polymersome formation? A clear description is recommended. 

After having reviewed some literature (https://doi.org/10.1016/B978-0-444-53349-4.00131-X; https://doi.org/10.1016/B978-0-08-096701-1.00204-4) we have modified the first paragraph at section 2 to focus on the main aim of this review (line 112). In addition, we also have modified Figure 2 (referred to as Figure 1 in the original document) to clarify the concept of self-assembly in the field of supramolecular polymers, thus now there is no polymersome figure, to direct the attention of the reader to supramolecular polymers, the focus of the review.

  1. The authors describe the mechanism of thermal switching of hydrophilic coils to hydrophobic globules in Figure 6. However, no relevant application-oriented examples were discussed. We encourage the team to discuss relevant application-oriented examples with the following information on nucleic acid delivery. For example, A thermoswitchable amphiphilic poly(2-n-propyl-2-oxazoline) (PnPrOx) was introduced between a hydrophilic poly(2-ethyl-2-oxazoline) block and a cationic poly(L-lysine) segment. The PnPrOx created a hydrophobic palisade due to the collapse of the PnPrOx segment above the lower critical solution temperature, resulting in a spatially aligned hydrophilic−hydrophobic double-protected polyion complex micelle loaded with plasmid DNA (pDNA). This hydrophobic palisade protected the pDNA from nuclease attacks and polyion exchange reactions, promoting effective gene transfection compared to micelle without hydrophobic palisade (Biomacromolecules. 2016;17(1):354-61).

We have included the suggested reference as a representative example in Section 4.1.1. (line 460).

  1. In the literature, we can find many reviews on stimuli-sensitive assemblies and their applications. How is this review significantly different from those existing reviews? An excellent review article is forward-looking and provides a framework for future direction for readers. 

There are several reviews in the literature about stimuli-sensitive assemblies, but none focused on responsive supramolecular polymers. There are several reviews focused on the diverse applications of functional supramolecular polymers (e.g. https://doi.org/10.1016/j.reactfunctpolym.2022.105209,  https://doi.org/10.1016/j.progpolymsci.2022.101635), but they are not mainly focused on the ability to respond to stimuli. Those publications cover a broad range of information about supramolecular polymers and their diverse applications. For this reason, we believe our review is quite different since we tried to provide information on supramolecular polymers and their biomedical applications from the point of view of responsivity.

  1. The authors discussed only the rationale for responsiveness to internal stimuli. However, examples were missing. We recommend the authors add essential applications for delivering therapeutic and diagnostic agents. For example, a thiol-responsive core crosslinking was introduced into the core compartment to prevent premature disassembly of nucleic acid therapeutics-loaded polyplex micelle, thereby promoting transfection efficiency. A pDNA-packaged crosslinked PEG-poly(L-lysine) (PEG-PLys) block copolymer-based polyplex micelle offered remarkable stability in the extracellular environment (10 µM glutathione), thereby enhancing cellular internalization at tumor site. After cytoplasmic entry, these crosslinks are cleaved in the reductive milieu in response to a highly reduced glutathione (10 mM) level, triggering the effective release of loaded pDNA ( 2014;35(20):5359-5368). Thiol core crosslinked micelles loading siRNA and mRNA effectively protected and delivered their nucleic acid cargoes into the cells (Macromol Rapid Commun 2022;43(12):e2100698, J Drug Target 2019;27(5-6):670-680, andMacromol Rapid Commun 2016;37(11):924-33).

We thank the reviewer for suggesting adding more examples. However, the ones suggested are not focused on supramolecular polymers. We have included one of the references (Biomaterials. 2014;35(20):5359-5368), in a brief mention of polyplex micelles and gene transfection, as explained in the next point (section 4.1.1., line 471).

  1. The authors discussed the rationale of pH-responsiveness. However, they failed to discuss the examples of pH-responsiveness in drug delivery. We recommend the authors incorporate the following discussion. MessengerRNA(mRNA) delivery has garnered spotlight attention as a versatile nucleic acid drug, not only for disease management and treatment but also for preventing infectious diseases (Proc Natl Acad Sci U S A2024;121(11):e2307800120). We recommend the authors introduce a few examples of stimuli-responsive polymers for effective intracellular mRNA delivery. For example, the delivery efficiency of mRNA delivery systems is greatly hampered by entrapment within endosomal/lysosomal compartments. To overcome this endosomal/lysosomal entrapment, an acidic pH-responsive negatively charged polymer (i.e., charge-conversion polymer) is coated onto a positively charged polyion complex. After reaching the endosomal/lysosomal compartment, followed by endocytosis, the coated charge-conversion polymer converts to a positively charged polymer from the original negative charge in the acidic pH milieu of endosomal/lysosomal compartments, facilitating smooth endosomal disruption. This design of charge-conversion polymer facilitated the efficient endosomal escape of the mRNA-packaged polyplex compared to a polyplex coated with a polymer lacking pH-responsiveness (Macromol Rapid Commun 2022;43(12):e2100754). 

We thank the reviewer for his/her suggestion, but again, our review is focused on supramolecular polymers. Although our review does not focused on polyplex micelles and gene transfection, we have included a brief paragraph in section 4.1.1. (lines 405-410) to include a reference here suggested (Biomacromolecules 2016, 17, 1, 354–361; https://doi.org/10.1021/acs.biomac.5b01456 ; ref 185).

  1. Line 256: matrix metalloproteinases (MMPSs). The correct acronym and most widely used is MMPs, not MMPSs. 

We thank the reviewer for observing this typo, we have changed the acronym in line 256.

  1. Lines 242 and 357: Glutathione (L–γ-glutamyl-L-cysteinylglycine, GSH). Use this amino acid description only at line 242. 

We thank the reviewer for observing this mistake, we have removed the amino acid description at line 357.

  1. Line 360: What is theranostic SP?

We thank the reviewer for raising this point. We have added the complete name of SP, supramolecular polymer, in line 360.

  1. Provide complete amino acids of codes used in GPLGIAGQ. 

The codes of amino acids from the suggested peptide have been provided (line 279).

  1. The manuscript mainly uses the previously published review articles as citations. We recommend the authors incorporate the original discussion and original article citations. 

We thank the reviewer for raising this point. We have added the original articles as much as possible, and we have reduced the number of review articles as much as possible. Changes in citations are highlighted in yellow in the ijms_revisions document. We would like to emphasize that we have cited 245 references, from which the vast majority are original articles and book chapters.

Round 2

Reviewer 1 Report

Comments and Suggestions for Authors

The authors may discuss more why the stimuli-responsive systems are important in the Perspective part. The most examples listed in this paper focused on triggering drug release by stimuli. However, more interestingly, advanced function may be achieved.

It is generally accepted that biological systems (e.g., protein) hold delicate structural hierarchies although how to build an ordered hierarchical structure is still not fully understood. The striking point is that most biological systems have structural rigidity on a certain spatial scale to maintain the stability of hierarchies though they still have the ability to change the structure for function through specific interaction, particularly triggered by environmental cues. Successful examples of nanomaterials/DDS should keep a good balance between stealth effect and interaction with diseased tissue. To break the trade-off between these two situations, dynamic modulation of the stealth effect through stimuli-responsive strategy may further advance the functionality and enhance the therapeutic efficacy. The authors may find some useful information on stealth and pseudo-stealth nanocarriers.

Author Response

We thank the reviewer for his/her interesting comments that will improve the quality of our review.

We have added new sentences in the Future Perspectives section on how more complex functions could be achieved (line 647) and how protein corona could be leveraged in the stimuli-responsive strategies to enhance their efficacy (line 619).

Reviewer 2 Report

Comments and Suggestions for Authors

No more comments. 

Author Response

Thank you very much for reviewing our manuscript.

Reviewer 3 Report

Comments and Suggestions for Authors

The authors satisfactorily addressed all the comments and suggestions raised by the reviewer. 

Author Response

(The authors gave the same response as above.)
